# The Impact of Political Stability on Environmental Quality in the Long Run: The Case of Turkey

**Dervis Kirikkaleli** *[ORCID] **and Aygün Osmanlı**

Faculty of Economic and Administrative Sciences, European University of Lefke, Lefke 99728, Turkey
* Correspondence: dkirikkaleli@eul.edu.tr

**Abstract:** In the 21st century, environmental problems are considered the biggest challenges. Turkey is one of the emerging markets that need to improve the quality of their environment. In the literature, how political risk affects the environment in Turkey has not been studied. In order to contribute to the issue, this study aims to determine the impact of political stability on the quality of the environment in Turkey. The present study investigates the effect of political stability on environmental quality, taking into account the critical role of economic growth, environmental regulation, patents in environmental technologies, and renewable energy consumption in Turkey from 1990 to 2019. The present study used nonlinear autoregressive distributed lag (NARDL) and dynamic ordinary least square (DOLS) models to investigate the effect of political stability on environmental quality in Turkey. The empirical findings show that political stability in Turkey reduces environmental deregulation by declining $CO_2$ emissions. Similarly, patents in environmental technologies and renewable energy consumption positively contribute to the environmental quality in Turkey by decreasing $CO_2$ emissions. On the other hand, economic growth has a significant positive effect on $CO_2$ emissions. This study suggests that political stability is an important indicator of environmental quality in Turkey. In order to ensure the continuity of Turkey's environmental sustainability, political tension in the country should be controlled by politicians, and it is recommended that Turkey should turn to and invest in renewable energy sources by following technological innovation.

**Keywords:** political stability; Turkey; environment quality; NARDL

## 1. Introduction

Among the biggest challenges in the 21st century are environmental issues. As an emerging market, Turkey needs to improve its environmental quality. In the literature, ways by which political risk affects the environment in Turkey have not been studied. To contribute to the issue, the impact of political stability on environmental quality in Turkey was studied. In the study, ways by which political stability, economic growth, renewable energy use, environmental regulation, and patents related to technology affect carbon emissions in the long term are explored by analyzing unique data for the case of Turkey. In addition, the literature on the subject is examined, and some recommendations are made according to the findings obtained within the scope of the study. In the study, NARDL and DOLS models are used to investigate the effect of political stability on environmental quality in Turkey.

Moreover, the present study answered the following questions: Does political stability affect environmental quality in Turkey in the long term? Does renewable energy consumption affect environmental quality in Turkey? Do environmental technology patents affect the environment in the long term? How does the growth of the economy in Turkey have an impact on the environment?

In a democratic process, the formation of an administrative structure that works effectively and efficiently in accordance with the rules of law is called political stability. The purpose of this definition is to see whether the decisions made by governments with

respect to political, economic, and social issues are suitable for the characteristics of the society, meet the expectations of the society, and meet their needs. Political and institutional factors are as important as economic factors with respect to the success of the economies of countries at the macroeconomic level in the 21st century, and researchers have published many articles supporting this view in the literature. Political stability is important for countries to develop economically. A favorable political environment's creation in the country is necessary for attracting foreign investors. However, having only a successful democratic regime, good governance, and a solid institutional structure is not enough to attract foreign investors. The sustainability of these features is another factor that should be considered in order to encourage foreign investors [1].

The concept of sustainability, which is evaluated in terms of its environmental aspect, is divided into three sub-dimensions: economic, social, and environmental sub-dimensions. Countries have their own sustainable development goals and implement various policies in order to achieve these goals. Although sustainable development policies in Turkey have only just begun, they have been put into effect on a micro-based basis. It can be said that the period of political stability that Turkey went through during this period had a positive effect on investments in the aforementioned sustainable policies [2].

Political stability defines a society and its ability to continue the path of development. It guides economic activities by making long-term predictions for decision makers who want to invest and making predictions in certain environments. Political stability is effective with respect to the environment, and it ensures the transfer of resources to future generations and their effective use. Political stability and environmental sustainability constantly influence each other. For the continuation of environmental sustainability, economic growth and ecological balance should be in harmony, and appropriate sustainable development policies should be prepared by taking the environmental factor into consideration when planning [3].

People have met the basic needs they need for their lives by making use of the ecosystem. The low population in the beginning allowed people to continue their lives by acting in harmony with nature without harming the environment. The place of the ecosystem in our lives is indispensable for the continuation of the life cycle, for meeting the needs of people, for their existence, and for the cycle of climate and natural events. As a result of the increasing population relative to industrialization, the raw material and energy needs of people have also increased, resulting in the increased use of raw materials and fossil resources in order to meet needs. The deterioration of the ecosystem by humans and the fact that nature is not allowed to renew itself negatively affected the environment and caused it to become polluted. Some of these effects are the increase in air temperatures, drought, decrease in precipitation, destruction of forests, increase in fires due to extreme heat, inability to obtain the desired efficiency in production, and the extinction of living things in the seas [4].

Rapidly increasing populations, energy needs, and industrialization have caused a decrease in raw materials and fossil resources due to increased use and the deterioration of the ecological balance. While using resources, the need for planning has emerged with respect to considering future generations. In order to reduce environmental pollution and waste, the use of nonrenewable energy sources should be reduced, and the use of renewable energy resources should be replaced; moreover, the country should be guided by producing policies in line with this direction. Planning that includes the environment in sustainable development plans can reduce environmental pollution so that nature is given the opportunity to renew itself due to the implementation of sustainable environmental policies. The ecological balance should be given importance in ensuring sustainable development, and the environment should be used effectively and efficiently in achieving the goals determined by economic growth; moreover, renewable energy should be preferred. Making investments that encourage the use of renewable energy resources in Turkey can increase the environmental quality [2].

Until the 1970s, states unconsciously consumed natural resources by thinking that they were unlimited and acted by focusing only on growth, damaging the environment. Studies focused on economic growth have caused an increase in environmental pollution and it has been understood that this situation is not sustainable. During the 1980s, it was understood that resources were not unlimited, and sustainable development models were created by including the environment in economic development plans. In addition, planning for environmental policies has been carried out with future generations in mind [5].

It is commonly reported in journalism that a government cannot survive because of political instability. In this view, uncertainty plays an important role. For the country to be considered unstable, a government does not have to fall. The government only needs to be seriously threatened by political tensions. It is possible to make this view functional, but it cannot be applied to real-world situations and is, therefore, only useful for anecdotal examples. For a government to remain stable, it must fulfill its stated duties and remain in power [6]. Clearly, governments that change frequently and have different preferences (often at odds) produce inconsistent policy outcomes. Moreover, the rapid turnaround indicates that the government cannot rely on sufficient political support. Political instability and economic growth are rarely examined empirically due to a lack of data. Nevertheless, some economists are interested in this relationship. According to Kuznets [7], much economic growth should not be expected due to unforeseen changes in the conditions of political turmoil.

Studies on environmental pollution have begun to pay attention to institutional factors recently. There can be direct and indirect effects on environmental quality from policies and regulations. To assess institutional quality, various proxies are used. In terms of governance structures that work effectively and efficiently, political stability, corruption control, and the rule of law are the most important indicators [8]. Studies investigating the impact on environmental quality have mostly focused on corruption and the rule of law as indicators of institutional quality [9]. Reduced carbon emissions are achieved via the formulation and regulation of environmental policies. In order to formulate and implement environmental policies, an impartial institutional system is crucial. It is crucial that a strong and stable government is in place in a country free of corruption in order to develop an effective environmental strategy and implement it. Nonetheless, if institutions are weak and there are flaws, firms will take advantage of pollution control protocols to maximize profits [10]. Additionally, the spatial institutional spillover effect is one way powerful institutions can reduce pollution domestically and spread the effect to neighboring countries. [11]. As economic growth proceeds, institutions that are impartial and effective contribute to reducing environmental pollution [12]. Environmental policies and green technologies cannot be formulated and regulated effectively due to weak institutions [13]. A country's ability to control environmental pollution depends on strong institutions. During the study's period from 2000 to 2016, political stability, corruption control, and the rule of law were compared to determine their effects on $CO_2$ emissions. It was observed that the rule of law, political stability, and corruption control all play a significant role in reducing carbon emissions and improving the environment. Observations indicate a decline in $CO_2$ emissions when political stability is compared to the rule of law [14]. Developing, regulating, and applying environmental laws without compromise are crucial to reducing $CO_2$ emissions. A major threat of the 21st century is climate change caused by carbon emissions and global warming [15]. To avoid the significant impacts of global warming and to limit the temperature increase by 2 degrees Celsius over the century, 196 countries signed the Paris Agreement on Climate Change in 2015 [16]. Environmental pollution policies such as the Paris Agreement and other related policies are largely determined by the quality of institutions [17].

Chowdhury et al. [18] conducted a study on the effects of foreign direct investments and other variables on the ecological footprint of 92 countries between 2001 and 2016. As a result of the research, the researchers, who observed that export and institutional quality reduce the ecological footprint, concluded that direct investment has a negative effect on

the ecological footprint. The ecological footprints of BRICS countries between 1995 and 2016 were examined by Kongbuamai et al. [19], and they examined economic growth, renewable energy consumption, nonrenewable energy consumption, industrialization, and the consistency of environmental policy. Based on their research, they concluded that economic growth, energy consumption from renewable and nonrenewable sources, and industrial activities have a direct and positive effect on the ecological footprint; however, consistency in environmental policies adversely impacts it. As a result, they concluded that BRICS countries should focus on preventive environmental policies in order to be successful in the field of environmental sustainability.

## 2. Literature Review

Acting with the idea that natural resources are unlimited until the 1970s, the environment was not given importance, and in 1980, plans for economic growth were made by including the environment in sustainable development plans. When approaching the year 2000, it was understood that the ecological balance of nature must be protected in order to meet increasing energy needs due to the industrial revolution. The aim of this study is to determine the effect of political stability on environmental quality in the long run. In the study, political stability and other variables are also included. Other variables used for this purpose are economic growth, renewable energy consumption, and environmental technology patents. The effect of each variable on the environment is determined, and its effect on $CO_2$ emissions is determined.

### 2.1. Political Stability and Environment

Yeşilay [20] aimed to explore Turkey's economic sustainability and concluded that Turkey is in a weak sustainable situation. The economic crises that have occurred in the world and the political stability that could not be achieved with the coalition governments have caused a great contraction in the Turkish economy. Turkey's trade and financial relations with countries are directly affected by the crisis and have led to serious demand reductions both at home and abroad, and negativities and uncertainties are expected from the economy in the future; accordingly, they have tended to decrease with respect to growth and employment rates.

Nathaniel et al. [21], in their studies conducted within the scope of N11 countries covering the period of 1990–2016, explored the effect of environmental regulations on ecological footprints and determined that the laws were not enough for reducing ecological footprints in the negative direction. Olcay [22] aimed to capture the sustainability performance of 24 countries covering the period of 1992–2012 and found that Turkey's economic values and human capital dimension were unstable and below expectations.

According to Purcel [23], low- and middle-income countries' political stability and carbon dioxide emissions are related, and upon reaching a threshold level, political stability can reduce pollution. Political instability can negatively impact pollution reduction practices by reducing any future policymakers' work, increasing uncertainty, and making policy changes more frequent [24]. In their study, Rizk and Slimane [25] examined the relationship between poverty and environmental degradation using global panel data from 146 countries between 1996 and 2014. The outcomes of Rizk and Slimane [25] reveal that the nonlinear relationship between poverty and carbon emissions leads to a further increase in poverty and environmental degradation. However, the resulting increase in institutional quality leads to a reduction in poverty and greater protection of the environment.

Studies in the literature prove that the world's leading democratic regimes are behind in the field of environmental protection. Battig and Bernauer [26] argue that this is because of the freedom of people in democracies, especially in the transportation sector. A problem in the functioning of the state causes more frequent policy changes and increased imbalance, negatively affecting pollution reduction practices [24]. Muhammad and Long [14] showed that in order to reduce carbon dioxide emissions and improve economic development, institutional factors such as political stability and corruption control play a very significant

role. $CO_2$ emissions are reduced in countries with low and high incomes but increased in countries with middle–low incomes due to trade openness. In a study conducted by Asongu and Odhiambo [27], for the case of 44 African countries, it was found that good management completely reduces the negative effects of $CO_2$ emissions. Sohail et al. [28], using the traditional ARDL model, observed that political stability reduces damage inflicted on the environment by reducing $CO_2$ emissions in the long run. When the non-traditional ARDL model is used, it has been observed that political instability also harms the environment in the long run.

### 2.2. Economics Growth and Environment

The Environmental Kuznets Curve (EKC) hypothesis underlines that there is an inverted U-shaped relationship between economic growth and environmental degradation. Yurttagüler and Kutlu [29] investigated the relationship between economic growth and environmental pollution within the scope of the EKC hypothesis for Turkey. As a result of the research, it was concluded that there was a relationship between the variables, but the findings did not confirm the EKC hypothesis. Özdemir and Kübra [30] conducted a study by examining economic growth and environmental pollution in Turkey. In the analysis conducted by considering the period of 1980–2015 for Turkey in the study, it was observed that a relationship between variables was determined in the long term, but the findings supporting the EKC hypothesis could not be achieved. Tekbaş [31] investigated Turkey's foreign trade and import and export variables within the period of 1970–2014, and they concluded that Turkey's exports, imports, and foreign trade increased environmental pollution in the country. Li and Lin [32] and Sadorsky [33] found evidence of a positive monotonous effect of income on environmental degradation across economies.

Ulucak and Erdem [34] used panel data analysis to identify how the ecosystem is handled in economic development models, and they explained that the gross national income of developing societies is more affected by the ecological impact, and the costs of environmental policies in these countries are higher.

Saqib and Benhmad [35] conducted a study to test how economic growth, energy consumption, and the population growth of 22 European countries affected the ecological footprint between 1995 and 2015. They concluded that there is a unidirectional cause-and-effect relationship from GDP to ecological footprint, and when compared to the intense energy consumption of the increasing population in these 22 countries, the relationship does not pose a serious problem.

The link between income and environmental degradation was first proposed by Grossman and Krueger [36]. The Environmental Kuznets Curve, named for its inverted U-shaped relationship between income and environmental degradation, was discovered in their groundbreaking study. Environmental degradation has been linked to $CO_2$ emissions in other studies. Chang [37] found that energy use and $CO_2$ emissions are both affected by financial development. Solarin et al. [38] determined the effects of economic growth, urbanization, and FDI in the long term on the ecological footprint in Nigeria. In this study, while economic growth pollutes the environment in a short period, it has a positive effect on environmental quality in the long run. While no negative effect of urbanization was detected, they determined that foreign direct investment and trade contributed negatively to the environment in the long run.

Ziaei [39] used the panel technique to identify the relationship between finance and the environment in European, East Asian, and Oceanian countries. The results showed that the development in the stock market increased $CO_2$ emissions. Xiong and Qi [40] aimed to explore the effect of financial development on the environment in the provinces of China, and they concluded that financial development reduces $CO_2$ emissions.

Yurtkuran [41] has conducted a study to determine the ecological footprint of newly industrialized countries, which covers the N11 countries for the period of 1971–2016. In the study, it was revealed that there is convergence in N11 countries while determining stagnation in the data of Indonesia, Philippines, and Pakistan, and because it was deter-

mined that environmental pollution is permanent in Turkey and some countries, it was recommended that the regulations of environmental policies should be reviewed by the politicians of these countries.

Aydın [42] examined the impact of environmental tax revenues on the environmental problems of selected OECD countries. As a result of the study, it has been determined that while Germany, Denmark, and Sweden benefitted from environmental taxes effectively with respect to solving their environmental problems, the situation is not the same for France and Italy. It was revealed that these two countries do not appropriately use the revenues they collect with respect to the environment.

### 2.3. Renewable Energy and Environment

The research conducted by Ayvaz and Över [43] investigated carbon emissions, technological developments, and the impact of renewable energy sources on economic development by using annual data for the G7 countries. Their findings reveal that while renewable energy and technological developments reduce economic expansion, the increase in carbon emissions positively affects economic growth. Kirikkaleli et al. [44] explored the impact of renewable energy consumption, technological innovation, and economic growth on the environment in China and their findings reveal that technological innovation and the use of renewable energy reduce $CO_2$ emissions while economic growth increases the amount of carbon.

Doğan and Şeker [45] confirmed that financial development has a negative impact on $CO_2$ emissions in 23 countries that use the most renewable energy in the world. The relationship between finance and the environment was, however, ignored in this study. The problem was also addressed by Alper and Onur [46] using Chinese data. The study found that financial development reduces $CO_2$ emissions from liquid fuel combustion and domestic, commercial, and public services. Gökhan et al. [47] examined how emissions trading and carbon emissions affect firm value in their research study. As a result of the evaluation, it was determined that companies that choose to benefit from renewable energy sources instead of fossil fuels can maintain their competitive advantage in the field of green capital by supporting the marketing mentality of society.

In the research conducted by Oğuz [2], it was concluded that if there is an increase in real GDP and political stability, it will cause an increase in the use of renewable energy resources. Murshed et al. [48] investigated how foreign direct investments affected renewable energy and environmental sustainability in Bangladesh during the 1972–2015 period. As a result of their analysis, they concluded that foreign direct investments increase the percentage of renewable electricity generation in the country's total electricity production levels but decrease environmental quality because they also increase the country's ecological footprint. This research study has revealed that the EKC hypothesis is valid for Bangladesh and that although foreign direct investment supports renewable electricity generation, it also seriously pollutes the country.

From the study of Uzar [49], it is important that renewable energy is backed by strong institutions if a reduction in pollution over the long term is to be achieved. The study stated that institutions play an important role in determining the quality of the environment, including the level of $CO_2$ emissions. It was determined that the stronger the institution, the more power a country has with respect to controlling $CO_2$ emissions because strong institutions can overcome problems such as corruption, financial problems, mismanagement, and bureaucratic incompetence. Goel et al. [50] investigated the direct effects of corruption and shadow economy on $CO_2$ emissions in 100 countries, especially in MENA countries. In their research, they found that $CO_2$ emissions were lower in countries with corruption and more shadow economies but higher in MENA countries. Adams et al. [51] investigated the long-term direct and indirect effects of democracy on environmental degradation in Ghana. They found that democracy directly reduces environmental degradation and indirectly minimizes the effects of urbanization. Osabuohien et al. [52] investigated international cooperation for good governance, electricity consumption, and $CO_2$ emissions relative

to measurements based on average corruption, the rule of law, and state competence in 27 African countries. They found that while the direct effect of good management is positive, it indirectly supports the $CO_2$-reducing effect of trade and electricity use.

### 2.4. Patent on Environmental Technologies, Regulation, and Environment

Demir et al. [53] examined whether technological factors reduce carbon emissions in Turkey. As a result of the examination, they concluded that domestic innovations increased the $CO_2$ level in Turkey in the early stages of economic development but that the increase in domestic innovations with development in the following periods reduced the $CO_2$ level. According to the results obtained, they argued that developed and developing countries could reduce their carbon emission levels by focusing on innovations. The same results were found by Cheng et al. [54] for OECD countries, and it was concluded that technological innovation provides a reduction in carbon emissions.

Shahbaz et al. [55] investigated the effect of technological innovations on the carbon level in China, and as a result of the study, they found that technological innovations reduced the $CO_2$ level. In this context, when they examined the literature, they concluded that technological innovations reduce carbon emissions and reduce environmental pollution.

Temelli and Şahin [56] investigated the impact of financial and technological development on the environment in 10 emerging market economies in their study within the period of 1995–2014. While no significant relationship was found between financial and technological development and carbon emissions in emerging markets, it was concluded that there was a positive and significant relationship between economic growth and $CO_2$.

Song [57] underlined that progress in green technology significantly increases an enterprise's total factor productivity via unit labor productivity improvements, and environmental regulation "forces" enterprises to implement progression in green technology. In order to enhance regional ecological well-being, environmental regulation is a key transmission path for the digital economy [58].

## 3. Data and Methodology

The present study investigates the effect of political stability on environmental quality, taking into account the critical role of economic growth, patents in environmental technologies, environmental regulation, and renewable energy consumption in Turkey from 1990 to 2019. The estimated model of the present study is shown below:

$$LCO_2 = f(LPATENT, LPRI, LGDP, LREC, ER) \qquad (1)$$

Equation (1) highlights that $CO_2$ emissions are determined by a combination of variables such as LPATENT, LPRI, LGDP, ER, and LREC. This model is converted into their regression form for empirical analysis as follows:

$$LCO_{2t} = \beta_1 LPATENT_t + \beta_2 LPRI_t + \beta_3 LGDP_t + \beta_4 LREC_t + + \beta_4 ER_t \varepsilon_t \qquad (2)$$

where $LCO_2$, LPATENT, LPRI, LREC, LGDP, and ER stand for production-based $CO_2$ emissions, patents on environmental technologies, political risk index, renewable energy consumption, gross domestic product, and environmental regulation, respectively. The Regulation on Environmental Impact Assessment came into force on 25 November 2014 while the Regulation on Environmental Permits and Licenses came into force on 10 September 2014. Therefore, the present study used a dummy variable to capture the effect of environmental regulations on environmental sustainability in Turkey, as advised by [57–59]. The present study used NARDL and DOLS models to investigate political stability on environmental quality in Turkey. To examine long-term equilibrium interactions between independent and dependent variables, the paper employs NARDL methods. NARDL approaches can address heteroscedasticity, serial correlation problems, and time series data in contrast to conventional ARDL methodologies. Moreover, although variables are integrated, the NARDL approach still applies. Finally, to obtain the nonlinear effect of

LPATENT, LPRI, LREC, ER, and LGDP on $LCO_2$ in Turkey, the present study employed the NARDL approach. Thus, the final expression of Equation (2), after considering the positive and negative shock of regressors using the NARDL model, is reported in Equation (3) as follows:

$$LCO_{2_t} = \beta_0 + \beta_1 LPATENT_t^+ + \beta_2 LPATENT_t^- + \beta_3 LPRI_t^+ + \beta_4 LPRI_t^- + \beta_5 LPRI_t^+ \\ + \beta_6 LPRI_t^- + \beta_7 LGDP_t^+ + \beta_8 LGDP_t^- + \beta_{10} ER_t + \varepsilon_t \tag{3}$$

In this study, the DOLS estimator is used as a robust test for the outcomes of NARDL. The DOLS eliminates serial correlation and endogeneity problems by using a parametric procedure.

Figure 1 and Table 1 illustrate the analysis flowchart and descriptive statistics of time series variables. As a result of the analysis of the data within the scope of the study, the graphs of political stability, economic growth, environmental technology patents, renewable energy consumption, and carbon emissions by year are presented in Figure 2. According to the pattern of the carbon emissions graph, although it decreased in some years, it generally continued until 2017 in an increasing manner and started to decline since that date. When we examine the political stability trend of Turkey with the data between research dates, as observed in the graphic above, it was determined that the stability in the country, which was governed by coalition governments in 1990–1991, was at a bad level and far below expectations. It has been determined that political stability increased in 1993 and decreased from 1994 to 1999, so it reached the worst level in research history in 1999. The transition to single-party rule in 2002 brought stability. It increased steadily until 2005, and stability was achieved via economic growth. It was determined that political stability reached the highest level in 2005 among the research dates. However, it was concluded that political stability went down in general with the effect of the world economic crisis between 2008 and 2012. In 2013, 8.5% growth was achieved in the country, and stability increased. However, in the following years, it was concluded that there was a slight decrease in the level of stability due to the political and economic developments in the country and in the world.

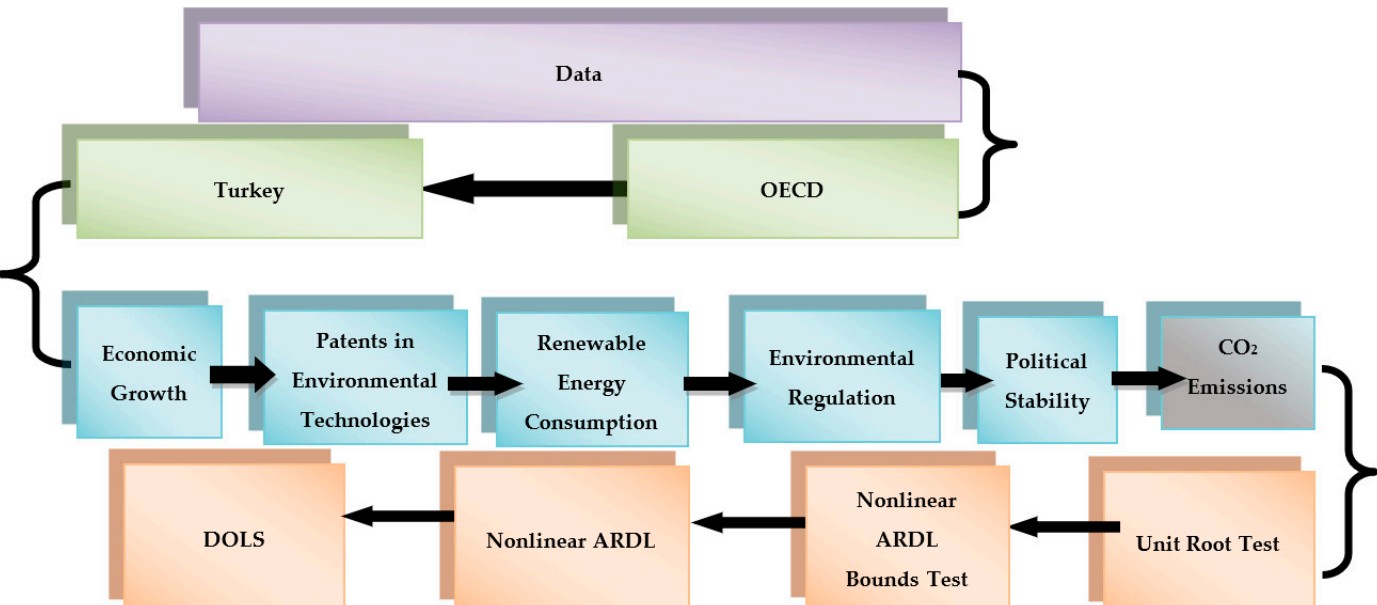

**Figure 1.** Analysis flowchart.

**Table 1.** Descriptive statistics.

|  | LCO$_2$ | LPATENT | LPRI | LGDP | LREC |
|---|---|---|---|---|---|
| Mean | 2.432585 | 0.880997 | 4.053574 | 3.894674 | 1.199090 |
| Median | 2.434903 | 0.886842 | 4.045954 | 3.898180 | 1.170850 |
| Max | 2.631356 | 1.391134 | 4.248495 | 4.079997 | 1.389222 |
| Min | 2.220599 | 0.511086 | 3.857215 | 3.734284 | 1.050802 |
| Std. Dev. | 0.122974 | 0.155301 | 0.092514 | 0.105623 | 0.103619 |
| Skewness | −0.037444 | 0.464126 | 0.248723 | 0.285966 | 0.360996 |
| Kurtosis | 1.763510 | 4.764836 | 2.351302 | 1.776231 | 1.772858 |
| J-B | 6.649565 | 17.23062 | 2.895796 | 7.907109 | 8.784321 |
| Prob. | 0.035980 | 0.000181 | 0.235064 | 0.019186 | 0.012374 |

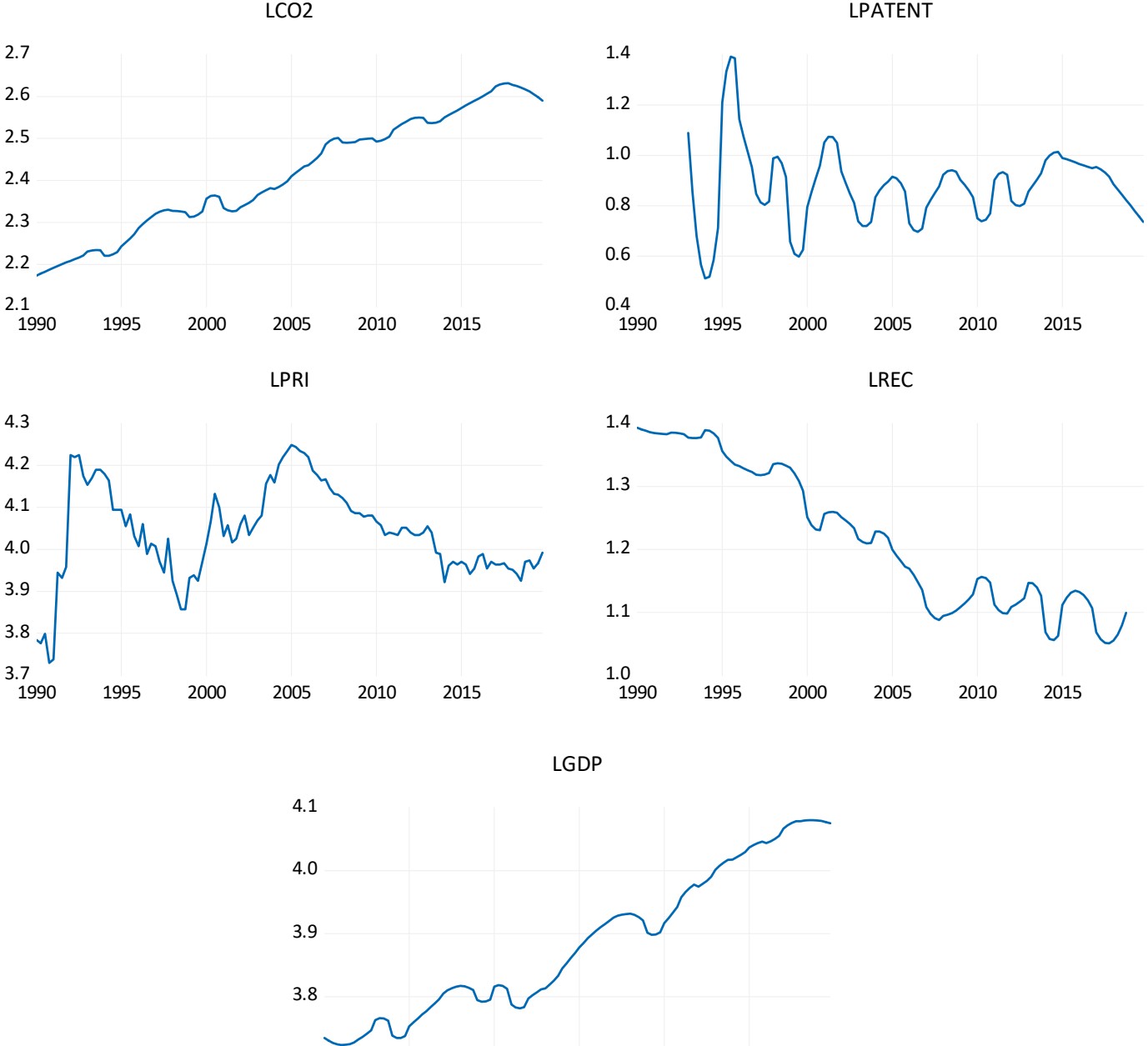

**Figure 2.** Time series variables.



## 4. Empirical Findings

The present study investigates the effect of political stability on environmental quality, taking into account the critical role of economic growth, patents in environmental technologies, and renewable energy consumption in Turkey from 1990 to 2019. Based on this aim, as an initial test, the BDS test was conducted to decide which estimate should be used to capture the aim of the study. By applying the unit root test, whether the data were stationary or not was examined. Since the nonlinear model is likely to give more accurate results, the nonlinear model has been used in the present study. Nonlinear ARDL long-run form values were calculated to determine the positive or negative effects of the effect. Finally, DOLS techniques were used as a robust estimator to support the outcomes of the NARDL test for the estimated model.

In Table 2, which test model would be used for performing the BDS test is decided. The test was applied for all variables. Results that are less than 0.01 indicate that each of our variables is independent, as reported in Table 2. For this reason, the nonlinear model was used in the study since the use of the nonlinear model is likely to provide more consistent results.

**Table 2.** BDS test.

| LCO$_2$ | | LPATENT | | LPRI | |
|---|---|---|---|---|---|
| **Dimension** | **BDS Stat.** | **Dimension** | **BDS Stat.** | **Dimension** | **BDS Stat.** |
| 2 | 0.198005 *** | 2 | 0.136643 *** | 2 | 0.158410*** |
| 3 | 0.332993 *** | 3 | 0.224677 *** | 3 | 0.265747 *** |
| 4 | 0.425665 *** | 4 | 0.283000 *** | 4 | 0.332975 *** |
| 5 | 0.489788 *** | 5 | 0.318450 *** | 5 | 0.379940 *** |
| 6 | 0.535707 *** | 6 | 0.335417 *** | 6 | 0.405961 *** |

| | LGDP | | LREC | |
|---|---|---|---|---|
| | **Dimension** | **BDS Stat.** | **Dimension** | **BDS Stat.** |
| | 2 | 0.196699 *** | 2 | 0.197245 *** |
| | 3 | 0.331343 *** | 3 | 0.332030 *** |
| | 4 | 0.424008 *** | 4 | 0.423148 *** |
| | 5 | 0.488331 *** | 5 | 0.485615 *** |
| | 6 | 0.534142 *** | 6 | 0.529287 *** |

Note: "*** denotes statistically significant at the 1% level" (Kirikkaleli et al. [60]).

After the nonlinear model was determined, whether the data were stationary was evaluated by applying the unit root test, and it was observed that since all data were less than 0.01, it was not necessary to use the second difference because the first difference was stationary. This indicates that the time series variables exhibit an I(1) behavior, as shown in Table 3.

**Table 3.** Unit root tests (ADF—break point).

| | LCO$_2$ | LPATENT | LPRI | LGDP | LREC |
|---|---|---|---|---|---|
| | | | **At Level** | | |
| t-Statistic | −2.748 | −3.990 | −3.531 | −2.308 | −3.000 |
| Break Points | 2003Q4 | 1998Q1 | 1991Q1 | 2009Q2 | 1999Q1 |
| | | | **At the First Difference** | | |
| t-Statistic | −5.694 | −8.679 | −13.954 | −6.097 | −6.044 |
| Break Points | 1993Q1 | 1995Q1 | 1992Q1 | 1993Q1 | 1992Q1 |

In this study, the new-generation nonlinear ARDL bound test model was used. The long-term impact of our independents, namely political stability, patents on environmental technologies, renewable energy consumption, and economic growth, on the environment is explored. The NARDL bound estimator reveals that time series variables have a long-term effect according to the reliability numbers of 10%, 5%, 2.5%, and 1%, as reported in Table 4, but whether the effects are negative or positive on an individual basis has not been determined. Therefore, the NARDL test is applied, and the outcomes of the test are reported in Table 5.

**Table 4.** Nonlinear ARDL bound tests.

| Test Statistic | Value |
|---|---|
| F-statistic | 3.672741 ** |

Note: "** denotes statistically significant at the 5% level" (Kirikkaleli et al. [60]).

**Table 5.** Nonlinear ARDL results.

| Variable | Coef. | Std. Error | t-Stat. | Probability |
|---|---|---|---|---|
| LPATENT_POS | −0.008091 | 0.019541 | −0.414087 | 0.6799 |
| LPATENT_NEG | −0.007754 | 0.027330 | −0.283725 | 0.7773 |
| LPRI_POS | −0.142261 *** | 0.033679 | −4.223989 | 0.0001 |
| LPRI_NEG | −0.017266 | 0.069549 | −0.248252 | 0.8045 |
| LGDP_POS | 1.289157 *** | 0.203617 | 6.331289 | 0.0000 |
| LGDP_NEG | −0.759868 ** | 0.358132 | −2.121756 | 0.0368 |
| LREC_POS | −1.404314 *** | 0.251016 | −5.594518 | 0.0000 |
| LREC_NEG | −0.237261 * | 0.122557 | −1.935913 | 0.0562 |
| ER | 0.106043 *** | 0.025293 | 4.192571 | 0.0001 |
| C | 2.245351 *** | 0.016338 | 137.4300 | 0.0000 |
| CointEq(−1) | −0.182514 *** | 0.031944 | −5.713535 | 0.0000 |

Note: "*, **, and *** denote statistically significant at the 10%, 5%, and 1% levels, respectively" (Kirikkaleli et al. [60]).

The nonlinear model is used in Table 5, and the results show how each independent variable affects the environment in the long run. Since the probability of patents is higher than 0.05, its negative and positive effects on the environment do not matter. However, the coefficient is negative, indicating that investment in environmental technology improves the environment or limits environmental degradation. This outcome can support the outcomes reported by Cheng et al. [55] and Shahbaz et al. [56], which underlined that technological innovations reduced $CO_2$ levels.

Table 5 also reports that the positive shocks of political stability affect environmental degradation. Turkey's political stability reduces carbon emissions. In other words, as political stability in the country increases, environmental sustainability and environmental quality also increase according to the estimated results. Purcel [23] examined the relationship between political stability and carbon emission in low–middle-income countries, and Sohail et al. [28] also conducted a similar study for the case of Pakistan and concluded that political stability reduces carbon emissions by making a positive contribution to the environment in the long term. As shown in Table 5, the *p*-values of the economic growth coefficient are below 5% and positive. Positive economic growth increases environmental degradation. In other words, as the economy grows, environmental degradation increases and environmental sustainability decreases. The findings of Solarin et al. [38], support our empirical results since Solarin et al. [38] concluded that in Nigeria, economic growth will pollute the environment in the short term and have a positive effect on environmental quality in the long term.

The fact that renewable energy makes up less than 5% and is determined as negative indicates that it makes a positive contribution to the environment. Doğan and Şeker [45] support the findings of our research since they found similar findings for countries using the most amount of renewable energy in the world. As a result, when the data were examined, findings were obtained. While political stability and renewable energy contribute positively to environmental sustainability, the environment is polluted in Turkey as economic growth increases. In addition, it has been determined that patents on environmental technologies do not have a significant effect on environmental pollution

As supportive evidence, the present study used the DOLS approach (Table 6). The main findings of DOLS are as follows: Political stability, renewable energy, and environmental technology patents contribute to environmental sustainability and are environmentally friendly. However, as the economy grows, the environment in Turkey deteriorates. At a 5% significance level, environmental regulations do not significantly affect environmental sustainability in Turkey.

**Table 6.** Robust test.

| | DOLS | | | |
|---|---|---|---|---|
| **Variable** | **Coef.** | **Std. Error** | **t-Stat.** | **Prob.** |
| LPATENT | −0.092338 *** | 0.031570 | −2.924866 | 0.0054 |
| LPRI | −0.144835 *** | 0.027807 | −5.208578 | 0.0000 |
| LGDP | 0.593217 *** | 0.078908 | 7.517813 | 0.0000 |
| LREC | −0.537601 *** | 0.059122 | −9.093000 | 0.0000 |
| ER | 0.018591 * | 0.010191 | 1.824324 | 0.0749 |
| C | 1.437359 *** | 0.474501 | 3.029198 | 0.0041 |

Note: "* and *** denote statistically significant at the 10% and 1% levels, respectively" (Kirikkaleli et al. [60]).

## 5. Conclusions

As a result of the increasing population relative to industrialization, the raw material and energy needs of people have also increased, causing an increased use of raw materials and fossil fuels in order to meet the needs and deterioration of the ecological balance. The deterioration of the ecosystem and the failure of nature to renew itself adversely affected the environment and caused it to become polluted. In ensuring sustainable development, the three main components of sustainability, which are environmental protection, economic growth, and social dimension, work in harmony, increasing the potential to meet people's wishes and needs. Institutions play a role in the implementation of environmental policies via policies and regulations. In order to reduce environmental pollution and waste, the use of nonrenewable energy sources should be reduced, and the use of renewable energy sources should be preferred. In the case of investments in technology, the increase in domestic innovations with developments in the future can reduce $CO_2$ emissions. Compliance with sustainable environmental policies while making development plans can facilitate the establishment of an ecological balance, and nature can be given the opportunity to renew itself.

There are problems with environmental issues in Turkey. In order to identify these and improve the quality of the environment, how political risks affect the environment in Turkey was investigated. In order to contribute to the solution, the impact of political stability on the quality of the environment in Turkey was identified. This empirical study has contributed to the literature because it is the first study conducted within the context of Turkey, and four different variables have been used together. In the study, the long-term effects of political stability, economic growth, patents on environmental technologies, and renewable energy use on the environment were examined. The evaluation was carried out by making use of unique data for the case of Turkey. NARDL and DOLS models were used in the study. How each independent variable affects the environment in the long run was

evaluated. The outcome of the present study reveals that there was no negative or positive effect of patents on the environment. However, investment in environmental technology improves the environment or limits environmental degradation. Political stability affects environmental degradation. Turkey's political stability reduces carbon emissions, and as political stability increases, environmental sustainability and environmental quality are expected to increase. Economic growth increases environmental degradation. In other words, as the economy grows, environmental degradation increases and environmental sustainability decreases. The use of renewable energy provides a positive contribution to the environment by reducing carbon emissions.

As a result, it is expected that political stability and renewable energy consumption in Turkey will contribute positively to environmental sustainability in the long run, and that the environment will be polluted in the case of increased economic growth. In addition, patents will not have a significant effect on environmental pollution. Political stability, renewable energy, and environmental technology patents will contribute to environmental sustainability in an environmentally friendly manner, but the growth of the economy causes environmental degradation in Turkey. Based on these findings, the following policy recommendations were made. In order to ensure the continuity of Turkey's environmental sustainability, investigating renewable energy sources and determining those that can be applied in the country are recommended. Research can be carried out within the scope of international rules that should be followed in order to protect the environment while ensuring economic growth and determining the legal regulations needed. If we want a healthy, livable world and sustainable environment for future generations, starting from educational institutions, environmental awareness should be imparted upon young people by providing training by expert personnel in order to create environmental awareness. R & D studies for environmental technologies should be supported, and increasing the institutional quality of a country should be ensured by coordinating with public-non-governmental organizations with the aim of increasing environmental quality.

**Author Contributions:** Conceptualization, A.O.; methodology, D.K.; software, D.K.; validation, A.O.; formal analysis, A.O.; investigation; A.O., resources, A.O.; data curation, D.K.; writing—original draft preparation, A.O.; writing—review and editing D.K.; visualization, D.K.; supervision, D.K.; project administration, D.K. All authors have read and agreed to the published version of the manuscript.

**Funding:** This research received no external funding.

**Institutional Review Board Statement:** Not applicable.

**Informed Consent Statement:** Not applicable.

**Data Availability Statement:** The variables used in this paper are collected from the database of the World Bank and OECD.

**Conflicts of Interest:** The authors declare no conflict of interest.

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
