# Peer review of "The Impact of Political Stability on Environmental Quality in the Long Run: The Case of Turkey"

_sustainability, doi:10.3390/su15119056_

Round 1

Reviewer 1 Report

The paper titled "The Impact of Political Stability on Environmental Quality in The Long Run: The Case of Turkey" is reviewed. The article needs minor editing before accepting it.   

The authors should first define abbreviations before using them, especially in the abstract

Line 376-377, Gross domestic product placed two times. One of them should be deleted. 

Figure 1 looks ok, but there are differences in the font sizes. 

For the table, to increase readability, the authors might show significance levels with stars

It might be useful to proofread the manuscript since there is minor grammatical errors exist in the paper.

Author Response

Reviewer 1

The paper titled "The Impact of Political Stability on Environmental Quality in The Long Run: The Case of Turkey" is reviewed. The article needs minor editing before accepting it.   

  • The authors should first define abbreviations before using them, especially in the abstract

Reponses:  We defined the all abbreviations

  • Line 376-377, Gross domestic product placed two times. One of them should be deleted. 

Reponses:  As suggested, we fixed the paper based on this suggestion.

  • Figure 1 looks ok, but there are differences in the font sizes. 

Reponse: thanks a lot for this suggestion, we fix the font size problem in Figure 1.

  • For the table, to increase readability, the authors might show significance levels with stars

Reponse:As suggested, we report the significance levels with stars in the tables.

  • It might be useful to proofread the manuscript since there is minor grammatical errors exist in the paper.

Reponse:As suggested, we proofread the manuscript before submitting revised version

Reviewer 2 Report

The study investigates the effect of political stability on environmental quality, taking into account the critical role of economic growth, patents in environmental technologies, and renewable energy consumption in Turkey from 1990 to 2019. The present study used NARDL and DOLS models to investigate the effect of political stability on environmental quality in Turkey. Overall, the study is interesting, and I have the following suggestions for the author's reference.

1.        The focus of the abstract needs to be further condensed, and attention should be paid to what is found in this study.

2.        It is suggested that the author clarify the innovation and the uniqueness of this research in the introduction.

3.        The article has formatting problems, please check carefully for revisions.

4.        There is a problem with the reference in line 41.

5.        The literature review should be refined and summarized, and the introduction of each literature should not be more than one sentence.

6.        Environmental regulation is an important factor affecting environmental quality(Song et al., 2022). If this factor is not taken into account, the results obtained will be biased. I suggest that authors refer to the articles listed and consider the impact of environmental regulations. Meanwhile,

7.        To ensure the robustness of the results, I suggest that the author adopt more robustness test methods, including but not limited to variable substitution and sample processing, etc., which can be referred to the literature listed(Alam et al., 2019).

8.        It is suggested that the author put forward some relevant policy suggestions in the conclusion part.

References:

https://doi.org/10.1016/j.resourpol.2022.102751;

https://doi.org/10.1016/j.eneco.2018.11.031;

Author Response

Reviewer 2

The study investigates the effect of political stability on environmental quality, taking into account the critical role of economic growth, patents in environmental technologies, and renewable energy consumption in Turkey from 1990 to 2019. The present study used NARDL and DOLS models to investigate the effect of political stability on environmental quality in Turkey. Overall, the study is interesting, and I have the following suggestions for the author's reference.

  1. The focus of the abstract needs to be further condensed, and attention should be paid to what is found in this study.

Response to Comment 1: The needed correction has been made as suggested.

  1. It is suggested that the author clarify the innovation and the uniqueness of this research in the introduction

Response to Comment 2: Necessary corrections have been made as suggested.

  1. The article has formatting problems, please check carefully for revisions.

Response to Comment 3: Based on your suggestion and based on the template of the journal, the formatting problem is eliminated.

  1. There is a problem with the reference in line 41.

Response to Comment 4: the all reference are revised and listed based on journal`s style. 

  1. The literature review should be refined and summarized, and the introduction of each literature should not be more than one sentence.

Reponse to Comment 5: Paragraphs containing the literatures were read and shortened into a briefer structure. Thanks for your suggestion!

  1. Environmental regulation is an important factor affecting environmental quality (Song et al., 2022). If this factor is not taken into account, the results obtained will be biased. I suggest that authors refer to the articles listed and consider the impact of environmental regulations.

Reponse to Comment 6: Dear reviewer thanks a lot for this suggestion, you are right for your suggestion but our estimated model cannot be performed with 5 independent variables, therefore we added this suggestion as a future study in the revised version

  1. To ensure the robustness of the results, I suggest that the author adopt more robustness test methods, including but not limited to variable substitution and sample processing, etc., which can be referred to the literature listed (Alam et al., 2019).

Reponse to Comment 7: Thanks a lot for this suggestion, we added DOLS outcomes as a robust test for the Nonlinear ARDL results.

  1. It is suggested that the author put forward some relevant policy suggestions in the conclusion part.

Reponse to Comment 8: As proposed, the conclusion section was developed and policy recommendations were made.

References:

https://doi.org/10.1016/j.resourpol.2022.102751;

https://doi.org/10.1016/j.eneco.2018.11.031.

Reviewer 3 Report

There is a need to improve the manuscript by taking the following steps:

1.     There is a need to explain the significance of this study in detail, especially the role of this research for further empirical research.

2.     The study has nicely explained the results and their interpretations. However, there is a need to explain the significance of the methodology (NARDL and DOLS). Why do the authors use this method?

3.     The authors should compare the current results with the previous studies. Try to highlight the contribution in terms of (data, methodology, and results). 

4.     Need to explain in more detail the facts and figures.

5.      In some parts of the section, Political Stability and Environment, authors should explain the literature review in more detail…..explain the reason. (e.g., (see lines 163-164) YeÅŸilay (2008) aims to explore Turkey's economic sustainability and conclude that Turkey is in a weak sustainable situation. [see line:185-190: In their study, Rizk and Slimane (2018) stated that environmental pollution can be reduced by political stability.It has been found that political stability has a huge impact on improving environmental quality in MENA economies by Al-Mulali, Thang, and Oztürk (2015). In addition, industry, commerce, urbanization and energy consumption also harm environmental quality.]

6.     Update the latest literature in section 2.3, especially the studies of the year 2022.

7. Improve the formatting, especially removing the unnecessary spaces in the text. 

8. The authors used an appropriate style of citations. See the author's guide.......Use [1], [2],...........

Author Response

Reviewer 3

There is a need to improve the manuscript by taking the following steps:

  1. There is a need to explain the significance of this study in detail, especially the role of this research for further empirical research.

Response to Comment 1: Thanks for the advice. The necessary changes have been made as suggested.

  1. The study has nicely explained the results and their interpretations. However, there is a need to explain the significance of the methodology (NARDL and DOLS).Why do the authors use this method?

Reponse to Comment 2: As requested, we added large paragraph to underline the significance of the methodology with their main advantages against others.

  1. The authors should compare the current results with the previous studies. Try to highlight the contribution in terms of (data, methodology, and results). 

Reponse to Comment 3: As suggested, the results were compared with previous studies.

  1. Need to explain in more detail the facts and figures.

Reponse to Comment 4 : Thanks a lot for this comment. The necessary changes have been made as suggested.

  1. In some parts of the section, Political Stability and Environment, authors should explain the literature review in more detail…..explain the reason. (e.g., (see lines 163-164) YeÅŸilay (2008) aims to explore Turkey's economic sustainability and conclude that Turkey is in a weak sustainable situation. [see line:185-190: In their study, Rizk and Slimane (2018) stated that environmental pollution can be reduced by political stability.It has been found that political stability has a huge impact on improving environmental quality in MENA economies by Al-Mulali, Thang, and Oztürk (2015). In addition, industry, commerce, urbanization and energy consumption also harm environmental quality.]

Response to Comment 5: Thank you for the advice. The fix has been made as suggested.

  1. Update the latest literature in section 2.3, especially the studies of the year 2022.

Response to Comment 6: as suggested we added new studies to the literature review section.

  1. Improve the formatting, especially removing the unnecessary spaces in the text.

Response to Comment 7: We organized our study based on the journal format.

  1. The authors used an appropriate style of citations. See the author's guide.......Use [1], [2],..........

Response to Comment 8: We organized our references based on the journal format.

Reviewer 4 Report

There are various flaws in the manuscript ID: sustainability-2280667. Counting and highlighting every flaw is hard; however, I pointed it out for the author's guidance. 

1. The abstract indicates nothing unique or surprising. For instance, political stability has a positive impact on the environment which everyone knows. I do not know what is for research in such findings. The authors need to translate these relationships into major findings by logical reasoning. rewrite the abstract clearly.

2. The intro section is very poorly started. I will say that looking at the first paragraph, even no one would like to read this article any further. Please consult with a good professor or researcher and learn how to write an intro. Besides, the intro section failed to let us know why this research is important. Thousands of studies are available on political stability and the environment.  

2. What is the research gap? This manuscript failed to let the audience know about the literature gap. What are the questions that this manuscript addresses?

3. How does this research contribute to the current streams of knowledge? How this research is unique and theoretical contributions. 

4. The literature review is not properly articulated and fails to identify what was done before and what is missing. A literature review is a critical review of previous literature that focuses on identifying what is not researched that should be investigated.

5. The methodology is not well-defined. Please, there are hundreds of papers in sustainability that focus on ARDL and NARDL economic modeling, read some of these articles and rewrite the method section. 

6. The results must be presented in APA style. 

7. there is no discussion section, and the conclusion is very poorly written. 

Author Response

Reviewer 4

There are various flaws in the manuscript ID: sustainability-2280667. Counting and highlighting every flaw is hard; however, I pointed it out for the author's guidance. 

  1. The abstract indicates nothing unique or surprising. For instance, political stability has a positive impact on the environment which everyone knows. I do not know what is for research in such findings. The authors need to translate these relationships into major findings by logical reasoning. rewrite the abstract clearly.

Response to Comment 1: Based on your suggestion, we re-write the abstract and added additional sentences.  

2.The intro section is very poorly started. I will say that looking at the first paragraph, even no one would like to read this article any further. Please consult with a good professor or researcher and learn how to write an intro. Besides, the intro section failed to let us know why this research is important. Thousands of studies are available on political stability and the environment.  

Response to Comment 2: Thanks for the advice. The necessary changes have been made as suggested.

2.What is the research gap? This manuscript failed to let the audience know about the literature gap. What are the questions that this manuscript addresses?

Response to Comment 2: as suggested we added additional sentences and also main research questions of the present study by underlining the research gap in the introduction section

3.How does this research contribute to the current streams of knowledge? How this research is unique and theoretical contributions.

Response to Comment 3: Thanks a lot for this comment. The necessary changes have been made as suggested.

4.The literature review is not properly articulated and fails to identify what was done before and what is missing. A literature review is a critical review of previous literature that focuses on identifying what is not researched that should be investigated.

Response to Comment 4: the majority of the literature review section is revised and re-written with new studies. 

  1. The methodology is not well-defined. Please, there are hundreds of papers in sustainability that focus on ARDL and NARDL economic modeling, read some of these articles and rewrite the method section. 

Reponse to Comment 5: As requested, we added large paragraph to underline the significance of the methodology with their main advantages against others.

  1. The results must be presented in APA style. 

Reponse to Comment 6: we organized the study in the revised version based on the template of sustainability journal. Thanks a lot for this suggestion

  1. there is no discussion section, and the conclusion is very poorly written

Response to Comment 7: The conclusion section was developed and suggestions were made for the findings. In addition, we added additional sentences for each findings of the study by comparing with previous studies.

Reviewer 5 Report

he material of the article is a good confirmation of the Kuznets Curve theory. And I believe that if the article is rewritten from the point of view of the argumentation of this theory using the example of Turkey, then the work will look more academic.

Author Response

Reviewer 5

The material of the article is a good confirmation of the Kuznets Curve theory. And I believe that if the article is rewritten from the point of view of the argumentation of this theory using the example of Turkey, then the work will look more academic.

Response to Comment 1: The studies conducted in Turkey within the scope of EKC are included in the article and various examples are given. However, we could not add the EKC hypothesis to our model due to the limited number of independent variable restrictions in the NARDL model. The estimated model did not work with 5 independent variables.

Round 2

Reviewer 2 Report

As I mentioned in the previous round of review, Environmental regulation is an important factor affecting environmental quality (Song et al., 2022). If this factor is not taken into account, the results obtained will be biased. I suggest that authors refer to the articles listed and consider the impact of environmental regulations. In addition, literature reviews is still relatively fragmented and requires further standardization.

Author Response

Response to the comment

Dear reviewer thanks a lot for your suggestion as suggested we added environmental regulation to the model to avoid the possibility of biased outcomes. As expected the signs of the coefficients did not change in the new results. We added 2-3 studies from (Song et al., 2022) as suggested. At the same time literature review is updated in the revised version.

Round 3

Reviewer 2 Report

Accept.